# Automated Training of Learned Database Components with Generative AI

Angjela Davitkova
RPTU Kaiserslautern-Landau
Kaiserslautern, Germany
angjela.davitkova@cs.rptu.de

Sebastian Michel
RPTU Kaiserslautern-Landau
Kaiserslautern, Germany
sebastian.michel@cs.rptu.de

## Abstract

The use of deep learning for database optimization has gained significant traction, offering improvements in indexing, cardinality estimation, and query optimization. However, acquiring high-quality training data remains a significant challenge. This paper explores the possibility of using generative models, such as GPT, to synthesize training data for learned database components. We present an initial feasibility study investigating their ability to produce realistic query distributions and execution plans for database workloads. Additionally, we discuss key challenges, such as data scalability and labeling, along with potential solutions. The initial results suggest that generative models can effectively augment training datasets, improving the adaptability of learned database techniques.

**ACM Reference Format:**

Angjela Davitkova and Sebastian Michel. 2025. Automated Training of Learned Database Components with Generative AI. In *Novel Optimizations for Visionary AI Systems (NOVAS '25), June 22–27, 2025, Berlin, Germany.* ACM, New York, NY, USA, 5 pages. https://doi.org/10.1145/3735079.3735322

## 1 Introduction

The increasing volume and complexity of data in modern database systems necessitate continuous optimization of indexing, query processing, and cardinality estimation. Traditional methods for database optimization rely on hand-crafted heuristics and rule-based techniques and often struggle under evolving workloads and large-scale datasets. Recent advancements in machine learning (ML) have introduced learned models that can improve database performance by predicting data distributions for cardinality estimation [12, 17, 22], optimizing query execution plans [16, 20], and enhancing indexing mechanisms [4, 13]. Despite their promise, learned database components, in particular query models, require extensive labeled training data to function effectively, as they rely on learning to identify patterns and generalize to different datasets. However, acquiring such data poses significant challenges.

First, there is a **Lack of Real-World Data.** High-quality training datasets consisting of key distributions, query workloads, and execution statistics are difficult to obtain due to privacy concerns and limited access to production database logs. Additionally, learned

models requiring queries for training face the **cold start problem**, where they struggle to make accurate predictions in the absence of sufficient initial training data, leading to poor performance until enough representative samples are available.

Second, problems arise in the presence of **Data Bias and Distribution Shifts.** Training on static datasets often results in overfitting, where models learn patterns specific to the training data but fail to generalize to unseen and evolving workloads. This limitation reduces their adaptability, leading to suboptimal performance when encountering real-world variations, such as workload shifts, seasonal trends, or dynamically changing database contents.

Third, **Scalability Issues** are an additional problem. Generating and labeling large-scale datasets manually or through query execution is highly computationally expensive, requiring substantial processing resources and time. The challenge is further amplified in dynamic database environments where both data distributions and query workloads evolve continuously, necessitating frequent regeneration and relabeling of training data. This overhead makes it impractical for applications, especially when aiming to maintain accurate and up-to-date learned models for indexing, cardinality estimation, and query optimization.

**To overcome these challenges, this paper investigates the feasibility of leveraging generative models to automate and enhance the process of training data generation for learned database components, specifically cardinality estimation and query optimization.** We focus on how well generative models can produce diverse and representative workloads and identify key aspects associated with synthetic data generation. We show that models such as GPT offer a promising alternative by using data semantics to generate meaningful workloads, beyond basic rule-based query generation. Furthermore, we address the limitations of the model, in particular, when considering selectivity-related tasks, and outline potential future ideas that can be investigated to enhance the adaptability and robustness of GPT, ultimately contributing to more efficient and accurate database optimization strategies. The data generation also provides an avenue for testing and benchmarking, facilitating the design of more robust components.

We present the requirements of workload synthesis for cardinality estimation and query optimization in Section 2, and discuss the potential of GPT in doing so in a first feasibility study in Section 3, before discussing related work in Section 4 and giving a brief conclusion and outlook in Section 5.

## 2 Methodology

### 2.1 Cardinality Estimation

Deep learning methods have shown promise in predicting query selectivity, enabling the selection of efficient execution plans to

improve performance. However, these models require high-quality labeled datasets for training, testing, and continuous improvement. The complexity of this task lies in having meaningful queries and the need to account for various query patterns, data distributions, and predicate selectivities.

*2.1.1 Problem Formulation for Cardinality Estimation.* The selectivity $\sigma(Q, D)$ for a query $Q$ and database $D$ with a total number of tuples $|D|$ can be estimated as: $\sigma(Q, D) = \frac{|R(Q,D)|}{|D|}$ where $R(Q, D)$ is the set of tuples that satisfy the query $Q$.

*2.1.2 Training Data Generation.* The key aspects of generating synthetic training data for cardinality estimation involve creating diverse query workloads and assigning corresponding selectivities. **Step 1: Generating Diverse SQL Queries** that cover different types of workloads and queries with varying complexity is the first step in training a cardinality estimator. The generated queries, depending on the cardinality estimator being tested, may involve various operations, leading to different outcomes:

- Simple Queries: Queries with straightforward selection conditions and projections.
- Complex Joins: Queries involving multiple joins, which require estimating the cardinality of intermediate results.
- Aggregations: Queries involving group-by clauses, needing complex cardinality estimations due to data grouping.

**Step 2: Generating Accurate Labels** involves assigning selectivity or actual cardinality values to the generated queries. These values are typically derived from actual query execution on data or approximated through sampling techniques. Although GPT can generate query syntax and structure based on patterns learned from data, it cannot analyze the underlying data distributions in real-time. We will discuss the extent of the model's capabilities in the feasibility study (Section 3).

## 2.2 Query Optimization

With learned query optimizers gaining traction for handling complex and dynamic workloads, generating high-quality training data is essential for their effective training.

*2.2.1 Problem Formulation for Query Optimization.* Given a query $Q$ and a database $D$, the goal is to select a candidate execution plan $P$ that minimizes the cost $C(Q, P, D)$ of executing query $Q$, i.e., $P_{\text{opt}}(Q, D) = \arg\min_P C(Q, P, D)$.

*2.2.2 Training Data Generation.* The process of generating synthetic training data for query optimization can be broken down into generating diverse SQL queries, simulating multiple execution plans for those queries, and estimating costs. **Step 1: Generating Diverse SQL Queries** is crucial for training query optimizers. Unlike in cardinality estimation, this step typically requires more complex queries involving multiple joins and nested queries. **Step 2: Query-Plan Pair Generation** involves that for each generated SQL query $Q$, we need to simulate multiple candidate execution plans $(P_1, P_2, ..., P_n)$. These execution plans represent different strategies for executing the query, such as:

- Join Ordering: Different orders of the joins between tables.
- Join Methods: Hash joins, nested loop joins, and merge joins.

- Access Methods: Index scans, full table scans.

Simulating multiple candidate execution plans for a single query results in a rich training dataset useful for training query optimizers to predict the most efficient plan based on query characteristics. **Step 3: Generating Accurate Labels** is to assign estimated costs to each generated execution plan, considering the following:

- Index Scan vs. Table Scan: Cost difference based on whether an index is used or a full table scan is performed.
- Hash Join vs. Nested Loop Join: Estimation of the cost of different join methods based on query structure and data distribution.
- Other Factors: CPU costs, memory usage, and disk I/O.

Accurate cost estimation relies on precise cardinalities, and the usage of GPT to do so will be further discussed.

## 2.3 Challenges and Discussion

*2.3.1 Diversity of Generated Queries.* The ability to achieve diversity in SQL queries is essential, as it is critical for developing an effective generator. Based on the data given, GPT optimizes query generation and workload adaptation by leveraging various database characteristics and usage patterns. **Schema-Aware Generation** allows that given a schema, GPT constructs a diverse set of queries that adhere to its structure, ensuring syntactic and semantic correctness. **Context-Aware Generation** allows providing context-aware requirements, creating realistic query workloads that are tailored to specific use-case workloads. When specified, GPT produces a mix of queries from different workload categories (e.g., heavy aggregation queries vs. light lookup queries) to simulate a realistic distribution of queries that the cardinality estimator might encounter in practice. Context-aware query generation can also be beneficial when the model struggles with certain query types and requires additional examples of those queries to improve its performance. **Workload Expansion** allows that when provided with an example workload, the model refines and expands it by learning from existing query patterns relevant to the given domain. By providing even a small example of query logs, GPT can generate queries that closely resemble practical usage patterns, improving workload representativeness and query performance insights.

*2.3.2 Generating Complex Query Patterns.* Complex queries involving multiple joins or nested subqueries pose another challenge for generative models. Detailed prompts are needed to fine-tune GPT and guide it in the right direction. For instance, by explicitly showing a wide variety of query types, including edge cases with nested joins and subqueries, GPT can also adapt and generate such realistic and diverse query patterns.

*2.3.3 Query Plan Diversity.* A major challenge in query optimization is the diversity of execution plans for a single query. Different strategies can be used for joins, scans, and aggregations, and generating a comprehensive set of these strategies is difficult. To solve this, GPT is capable of generating multiple alternative plans for the same query. Further, these plans need to be validated against real-world query execution to ensure their diversity and realism. By fine-tuning the model on a wide range of plans from real execution logs, GPT can produce a broad set of execution plans.

*2.3.4 Selectivity & Cost Estimation.* By default, GPT cannot accurately perform counting or cardinality estimation because it operates purely based on patterns learned from large amounts of text data rather than actual data distribution or statistics. Since accurate cost estimation is dependent on cardinality estimates, the same problem is present for labeling in query optimization. However, the model can incorporate guidance for selectivity information to generate queries that reflect realistic database access patterns, that align with expected data distributions. For example, providing information such as table sizes, column distributions (e.g., distinct counts, value frequencies), and predicate selectivities, helps the model, to some extent, to generate realistic queries that balance low- and high-selectivity patterns.

*2.3.5 Adaptation to Different Databases.* Different database systems have varying syntax as well as optimizers and execution strategies. A query optimized for one system may not be optimal for another. GPT is adaptable to multiple database engines when properly instructed. Training data generation needs to be adapted to represent workloads suited for distinct database engines ensuring that learned models generalize across various systems. This adaptability allows the generated data to be more useful across multiple platforms, improving the generalization of query optimizers.

*2.3.6 Scaling to Large Datasets.* One key issue with generating large-scale training data is the computational cost of generating queries. A good generative model should be scalable, meaning it can efficiently generate large volumes of high-quality data. The query generation requires generating a defined number of *n* queries. Due to the model's output limitations, when *n* is large enough that a single response cannot accommodate all queries, we must generate them in batches to ensure the desired quantity is met. When considering query optimization, generating a large volume of diverse training data can become computationally expensive as the number of potential query plans grows. By introducing small variations to existing queries (e.g., modifying join conditions), GPT can produce more data from a small set of original queries. This helps in scaling the training dataset without requiring an overwhelming computational cost. While the generation of diverse queries can be efficiently parallelized, scalable labeling remains challenging and requires incorporating traditional indexing methods or cardinality estimators for a fast and accurate execution.

*2.3.7 Fidelity of Generated Data.* While GPT can generate syntactically valid SQL queries, guaranteeing they represent realistic use cases and query patterns is crucial for effective training. This challenge can be addressed by domain-specific fine-tuning of GPT on real-world query logs, ensuring that the generated queries reflect typical user behavior and database workloads. It would be helpful to detect **similarities between the generated data and real-world data** in terms of similar joins, filters, and subqueries, variations expressing the same intent, and deeper syntactic and semantic resemblances. **Domain expert reviews or user studies** can also be useful in assessing whether generated data reflects typical database behavior by manually inspecting a subset of queries, cardinalities, or execution plans. Finally, the **performance of a model** trained on generated data and evaluated using real-world data can act as a direct indicator of the generated data's quality.

## 3 Feasibility Study

For the first insights on the usefulness of generative AI to synthesize training data for learned database components, we give an overview of key metrics and first observations on how well GPT can handle the different tasks, using OpenAI's GPT-4o over a modified, simpler version of the IMDB schema and data [14].

### 3.1 Diversity of Generated Queries

*3.1.1 Schema-Aware Generation.* To analyze how GPT generates queries, we start by providing the IMDB schema and using a simple prompt to generate a diverse set of queries for it. The model generates various queries **primarily consisting of equality predicates, followed by range predicates, mixed predicates, and nested queries that use the IN operator.** While simpler queries with no or fewer joins are prioritized, more complex multi-table joins are also included. The tables and columns that are used vary in different queries, where the constraints given by the schema are used in the join predicates. During the generation, we can also restrict an equal number of queries for each predicate, which is particularly useful for classification or testing tasks.

*3.1.2 Context-Aware Generation.* The generation of queries using GPT also has the capability to be context-aware, e.g., different approaches might focus on different predicate conditions to showcase their performance. An example tailored prompt can be as follows:

```
Generate a workload of SQL queries with inequality
predicates for testing a cardinality estimator for
the IMDB dataset. The workload should include queries
with simple and compound predicate inequalities.
```

resulting in queries specifically tailored for this use case:

```
SELECT * FROM movies WHERE rating > 7.5;
SELECT * FROM movies WHERE release_year < 2000;
SELECT * FROM movies WHERE duration
BETWEEN 90 AND 150 AND rating >= 6;
```

A step further would be the creation of test queries mimicking a specific use case, such as:

```
Generate a workload of SQL queries for testing a
cardinality estimator for the IMDB dataset by
assuming the workload of an accountant of movies.
```

resulting in queries involving information relevant to an accountant, such as **revenue, movie budgets, durations, and ratings**.

*3.1.3 Workload Expansion.* Given a sample workload and insufficient data, we aim to generate additional queries to supplement it. For instance, if the previously represented inequality queries are provided as a sample, the following queries are generated:

```
SELECT * FROM movies
WHERE duration > 150 AND rating > 7.5;
SELECT * FROM movies WHERE release_year
BETWEEN 1980 AND 2000 AND rating >= 6.5;
```

The model extended the queries by varying the numeric ranges and combining conditions while maintaining the original intent.

### 3.2 Selectivity & Cost Estimation

Intuitively, when only the schema is provided, the model does not know anything about the column distribution, and it relies on

| Given to the Model | Equality Predicates Only | | Inequality Predicates Only | |
|---|---|---|---|---|
| | Non-Selective | Selective | Non-Selective | Selective |
| Boundaries & Schema | 0.02920 | 0.00330 | 0.70965 | 0.0382 |
| Sample & Schema | 0.0342 | 0.000679 | 0.6073 | 0.0133 |
| Histogram & Schema | 0.0328 | 0.0102 | 0.7143 | 0.0088 |

**Table 1: Avg. Selectivity for Different Query Types**

| # Queries | 10 | 20 | 30 | 40 | 50 | 100 |
|---|---|---|---|---|---|---|
| Avg. Time per Query (ms) | 486 | 377 | 380 | 415 | 309 | 310 |

**Table 2: Execution Time for Different Numbers of Queries**

assumptions. For instance, if we consider a car dataset when asked for a selective query, GPT invents values that are selective in a global sense and incorporates them in the query, e.g., "Rolls-Royce". Thus, it is crucial to provide additional information to the model.

To test the behavior of generation based on selectivity, we retrieve the starting year of the title table. We first create two broad prompts, including the sample of the array, one for highly selective and one for not that selective queries in the form:

```
Create 20 query predicates with high selectivity that
produce a small number of results as an outcome.
```

*The model generates queries based on the understanding that equality predicates are associated with high selectivity, while wide-range predicates are linked to low selectivity.* Since GPT cannot provide an accurate count of the individual elements in a sample list, the selectivity is assumed to be solely based on assumptions.

To further underpin the problem, we enforce it to generate selective and non-selective predicates for different settings (Table 1). **Boundaries & Schema:** GPT generates queries based on the minimum and maximum values of the data but lacks knowledge of the distribution of values within the specified range. As a result, it may either distribute queries evenly across the range or rely on heuristics to guess common values, such as assuming that recent years are more frequently represented. For equality queries, this often leads to poor selectivity since GPT does not have information on which values are more common or rare. In the case of inequality queries, GPT sets the predicate boundaries based on the asked selectivity. **Sample & Schema:** If the sample is representative, GPT can infer patterns, such as clusters of frequently occurring years. However, it does not count occurrences precisely; instead, it estimates based on the patterns it observes within the sample. As a result, its accuracy in determining selectivity may be limited. **Histogram & Schema:** Similar to previous cases, the model cannot keep track of the exact data but can recognize what is considered selective versus non-selective. While GPT may not be able to compute exact counts, it can make better estimates when provided with a histogram, as it has access to the frequency distribution. The histogram helps the model prioritize common values, but the estimates are still based on patterns rather than exact counts.

While GPT can distinguish between highly selective and non-selective queries, it cannot generate values in a pre-specified range. None of the approaches can provide reasonable results when asked to generate queries in a specific selectivity range, which poses the question, how will we label the data?

## 3.3 Scaling to Large Datasets

Last but not least, the runtime cost for generating synthetic workloads must also be acceptable. To obtain ballpark numbers, we asked GPT to generate a number of queries and measure the time it takes.

As we can see in Table 2, generating a larger number of queries slightly reduces the overall average time. The execution time also depends on factors such as the query size, the input schema, and the number of tables included in the queries.

## 4 Related Work

Benchmarking database systems is a core ingredient of scientific publication and industrial performance studies, with benchmarks like TPC-H [1], SSB [18], the join order benchmark (JOB) [14], or TPC-H Skew [3].

Hilprecht and Binnig [10] propose zero-shot learning to train learned components of a database without costly (re-)training required in former learned approaches [19]. While aiming at the same problem as we do, they do this from a completely orthogonal angle. They propose learning based on data and workloads, and a new benchmark for which they implement a workload generator using different patterns of queries. Differently, we propose to leave the generation of workloads to the LLM—we do not suggest any new learning methodology. Another related field is automated database tuning, with recent work on using machine learning methods to tune various aspects of a database, like index creation, buffer sizes, etc. [23], to use optimizer hints [21], or using (Large) Language Models to understand the meaning of parameters and follow their suggestion [2, 8]. None of these works specifically aims at training learned components inside a database system.

Essential to our goal of using LLMs for workload generation is their ability to understand database schemas and to generate SQL queries. Work on Text-to-SQL [5, 11] is now turning toward using LLMs [6, 9, 15] and recent off-the-shelf LLMs like GPT show already an impressive, human-like level of understanding schemas and query intent and perform very well in Text-to-SQL tasks [7].

## 5 Conclusion & Outlook

We proposed using generative AI to enhance the applicability of learned database components by automatically producing database workloads to support their development and optimization. Through an initial feasibility study, we highlighted key requirements and challenges and demonstrated that models like GPT can effectively assist in working with database schemas and SQL for query generation, when provided with well-designed prompts.

While GPT models are effective at generating diverse SQL queries under various constraints, generating queries that adhere to specific selectivity requirements remains a significant challenge. Achieving fine-grained control over selectivity requires more than language generation—it necessitates the integration of indexing structures to guide the generation for correct selectivity filtering and label assignment. Another promising direction is the generation of parameterized queries, where selectivity variations are introduced systematically based on additional statistics, offering greater control over the diversity and representativeness of generated workloads.

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
