# OpenReview forum: "Automated Training of Learned Database Components with Generative AI"
_ACM.org/SIGMOD/2025/Workshop/NOVAS — NOVAS 2025_

### Official Review · Reviewer_vbYH · 2025-04-14

**Confidence:** 4

**Improvement Opportunities:**

- The approach of using generative models to produce training data for learned indexes is not entirely convincing for two primary reasons:
First, the data acquisition cost: The paper does not fully address whether the reduction in optimization overhead justified the high cost associated with acquiring such data. It would be beneficial to consider the complete end-to-end cost of optimization, including both overhead and gains. Second, the labeling errors: Relying on LLMs to generate data may introduce labeling errors, which can negatively impact the performance of the learned models. More discussion is needed about how these errors might affect downstream tasks.

- The paper does not provide a thorough analysis of the overall gains from integrating learned indexes into databases. A more in-depth exploration of this potential benefit would strengthen the motivation for the proposed approach.

- In Section 3.3, the evaluation of GPT’s cost is based only on execution time. However, the computational cost is arguably more critical—especially since execution time can vary with model size. Furthermore, the settings for the experiments (as shown in Table 2) aren't clearly specified, which undermines the clarity of the analysis.

**Minor Comments:**

- It is unclear what role the execution plan plays in the evaluation and how the cost is defined. Does the cost include the expense of generating the dataset with the LLM? Clarification on these points would improve the discussion.
- The paper states that “GPT is more effective” without specifying what that means. Is the improvement measured in terms of accuracy, efficiency, or another metric? Additional details about the evaluation criteria are necessary.
- More details are needed about the overall approach. For example, when stating that “the model can incorporate guidance for selectivity information to generate queries that reflect realistic database access patterns,” it would be helpful to specify what kinds of guidance are being integrated—perhaps with an illustrative example.
- The use of parentheses around “IMDB” is unnecessary and should be removed for consistency.

**Short Summary:**

The paper explores the possibility of using generative models to synthesize training data for learned database components, and it outlines the requirements and challenges associated with this approach.

**Strong Points:**

- The writing is clear and accessible.
- The paper provides a detailed overview of the key requirements and challenges associated with generative models.

---

### Official Review · Reviewer_uju6 · 2025-04-19

**Confidence:** 4

**Improvement Opportunities:**

W1. The paper should articulate the problem definition better.  First, learned database components could be query models (e.g., https://arxiv.org/abs/1809.00677) or data models (e.g., https://arxiv.org/abs/1909.00607). Training the latter only needs access to the data while the former uses both query and data distribution. The cold start problem perhaps applies more to the latter category. In any case, the paper should actually define what "training data" is needed. For example, do we need to generate both data records for a schema, or do we only need SQL queries?

W2. The paper should discuss why using generate models are a good idea. Generating queries can be done without AI (e.g., https://www.vldb.org/archives/website/2004/protected/eProceedings/contents/pdf/IND2P3.PDF), and in fact there are simple ways of doing it (just go over columns and generate a simple SQL statement with a desired selectivity). I do think generative models can be good at predicting workload using data semantics, but this needs to be highlighted better in the paper.

W3. The paper should at least acknowledge that distribution of data generated by generative model and the real-world distribution can be different, and perhaps discuss potential directions to bridge the gap.

**Minor Comments:**

NA

**Short Summary:**

The paper studies the problem of generating training data for learned database components using generative models. It specifically considers generating data for cardinality estimation and query optimization problems, and discusses the challenges of using a generative model.  It performs a feasibility study and discusses the diversity, selectivity and scalability when generating the queries.

**Strong Points:**

S1. The idea of generating queries with generative models for training learned components is interesting.
S2. The paper includes a feasibility study
S3. Discussion on challenges is insightful

---

### Official Review · Reviewer_BkEg · 2025-04-24

**Confidence:** 3

**Improvement Opportunities:**

(W1) The vision should incorporate an evaluation of actual learned database components.

The authors focus on the question of whether generative language models are able to produce viable SQL queries when prompted. This seems like pretty low-hanging fruit that was probably done already. There are still many questions that need to be answered in order to make this vision truly viable. For example, how would we evaluate the quality of trained data? The gold standard would be to actually use it to train a learned component (e.g. a query optimizer) and see if the quality of the query plans improves as a result of the newly synthesized training data. I was not able to find an adequate discussion about this topic. Consequently, it is hard to assess the success criteria of this work and what steps would it need to take to demonstrate that the proposed approaches actually achieve their initial goal.

(W2) It is not fully clear how the proposed vision is able to overcome the challenges the authors laid out in the introduction.

For example, as far as I understood, the scalability issues come from having to run all workloads that are present in the training data against a real database in order to obtain the labels for training the learned components. However, in section 2.3.6 when the authors talk about scalability, they talk about the ability to generate many queries using the language model, which is not the same thing and the question of the cost of producing training labels remains open.

**Minor Comments:**

(D1) The authors mention "distribution matching metrics" but it is unclear how they would be applied to comparing the distributions of SQL queries.

(D2) It would be good if the authors gave us more details about their experiments. For example details about the exact GPT model that was used.

(D3) In the conclusion, the authors write: "We highlighted requirements and challenges and showed with a first feasibility study that it seems
plausible for models like GPT to be of great help to understand database schemas, SQL, and the ability to generate queries if the prompts are adequate." -- GPT can help us understand database schemas and SQL? This is a bit of an odd statement.

**Short Summary:**

The authors present a vision of using generative language models as a tool for synthesizing training data for learned database components. They identify several issues with developing state-of-the-art learned database components: (1) lack of real-world training data, often due to privacy concerns; (2) data bias and distribution shifts, when the distribution of the training data does not match the data distribution encountered at test time; and (3) scalability issues, resulting from the computational cost of executing every workload that is added to the training dataset. The learned components that the authors focus on in this work are cardinality estimators and query optimizers. They describe their specific approaches to data generation for these two scenarios, outlining their prompting strategies and discussing how they are used to overcome some of the challenges. They also experimentally validate that prompting a model to generate more selective queries for an example dataset (the authors use the IMDB dataset) indeed ends up generating more selective queries.

**Strong Points:**

(S1) The problem of insufficient high-quality data for training learned database components is important.

(S2) The approach of generating synthetic data to counteract this problem is a potentially viable approach.

(S3) The overall presentation is relatively easy to follow.